# Following Clues, Approaching the Truth: Explainable Micro-Video Rumor Detection via Chain-of-Thought Reasoning

## Abstract

The rapid spread of rumor content on online micro-video platforms poses significant threats to public health and safety. However, existing Micro-Video Rumor Detection (MVRD) methods are generally black-box, which lacks transparency and makes it difficult to understand the reasoning behind classification decisions. In this work, we introduce **ExMRD**, a novel **Ex**plainable **M**icro-video **R**umor **D**etection framework designed to generate detailed and coherent explanations for enhancing MVRD. Inspired by the powerful reasoning capacity of Chain-of-Thought (CoT), we introduce a novel inference mechanism called $R^3CoT$– consisting of *Refining*, *Retrieving*, and *Reasoning* on MVRD. This mechanism enables Multimodal Large Language Models (MLLMs) to reorganize the original video content, retrieve domain knowledge related to rumors, and generate explainable conclusions regarding whether the micro-video contains rumor information. Instead of directly fine-tuning MLLMs for MVRD, which is computationally expensive, we propose a Small Language Reviewer (SLReviewer), which distills the outputs of $R^3CoT$ guided MLLMs to ensure efficient and reliable predictions. Extensive experiments on three real-world benchmarks demonstrate that ExMRD significantly outperforms competitive baselines while providing high-quality rationales.

## Keywords

Micro-video rumor detection, explainability, chain-of-thought, multimodal large language models

## 1 Introduction

The exponential growth of online micro-video platforms such as TikTok, YouTube Shorts, and Snapchat has revolutionized information consumption worldwide [2, 21, 33]. With billions of active users, these platforms enable rapid creation and dissemination of micro-videos, offering unprecedented speed and reach in information sharing. However, this convenience comes with the proliferation of misinformation and rumors, which often evade scrutiny and fact-checking [4, 25, 54]. A striking example, as shown in Fig. 1, occurred during the COVID-19 pandemic when a TikTok video falsely claimed that injecting disinfectants could "kill" the virus. This misleading content amassed millions of views and resulted in a spike in accidental poisonings, highlighting the tangible harm caused by misinformation on micro-video platforms, and underscoring the urgent need for effective methods to detect and address rumors in micro-videos, a task *a.k.a.* Micro-Video Rumor Detection (MVRD).

Existing MVRD approaches primarily focus on utilizing multiple modalities – such as text, audio, video content, and social context – to improve detection accuracy [8, 17, 36, 37, 41, 42]. For example, FakingRec [8] analyzed the process of rumor creation by examining material selection and editing behaviors on micro-video platforms, while NEED [37] leveraged relationships between videos related

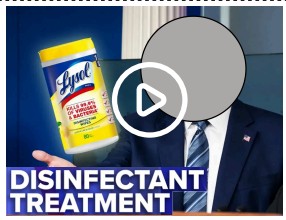

**Original Content:** Could be a secret to beating virus? You won't believe it! #MiracleCure #COVID19 #MustTry Now And then I see the disinfectant, ... And is there a way ... by injection inside or almost a cleaning? Because you see it gets in the lungs, .... So it'd be interesting to check that. So you're gonna have to use medical doctors with

**After Refining**: The video shows a speaker suggesting the **possibility of using disinfectants** internally, such as **through direct injection**, to combat COVID-19 viruses, in the lungs, and may require medical professionals assess the idea.

**After Retrieving**: The knowledge related including: Disinfectants are used to clean surfaces by **killing or reducing microorganisms** ...; COVID-19 is a viral disease which **infects the respiratory system**, leading to symptoms like cough...

**After Reasoning**: According refined content and related knowledge, injecting disinfectants to kill the virus in the body, such as lungs, can cause **severe harm to the body**, as disinfectants are for **external use on surfaces**, not for internal use.

Fig. 1: In this micro-video rumor, viewers are misled into believing that injecting disinfectants can kill the COVID-19 virus, leading to a rise in accidental poisoning incidents. Text in video: *Disinfectant Treatment.*

to the same event to improve rumor detection. Despite these advancements, current approaches often perform black-box detection, oversimplifying or overlooking the critical reasoning needed to provide explainable justifications for the final prediction. This lack of transparency makes it difficult for viewers to understand why a video is classified as a rumor, undermining trust and limiting the effectiveness of rumor mitigation strategies. In contrast, an explainable model is essential to enhance the effectiveness and trustworthiness of MVRD systems. Both users and platforms need to comprehend the specific factors that lead to a video being identified as a rumor or genuine content. However, developing such a model presents several critical challenges:

**C1: Inconsistent Video Quality and Misleading Metadata.** Micro-video platforms often contain content with inconsistent video quality and misleading metadata, which pose significant obstacles for interpretable rumor detection. Poor visuals and audio due to the varied skills and equipment of content creators make it difficult for detection models to capture critical cues. Pre-trained models optimized for high-quality data often perform poorly when applied to low-quality inputs, resulting in frequent misclassification [15, 32, 44]. For instance, the video tags in Fig. 1 (e.g., #MiracleCure, #MustTry) represent low-quality text content. These tags lack informative value and do not contribute meaningful content for the model to analyze. This issue arises because many video creators usually employ misleading titles and tags to attract attention, which may distort the model's interpretation of the actual content. These inconsistencies pose significant challenges to current

methods, further undermining the accuracy and trustworthiness of rumor detection models.

**C2: Lack of Domain Knowledge and Reasoning.** Effective rumor detection requires domain knowledge and logical reasoning to interpret complex content accurately. For example, identifying the micro-video in Fig. 1 as a rumor requires a basic understanding of COVID-19 biology and the proper use of disinfectants, which are intended for surface cleaning rather than for internal use. With this knowledge, and through logical reasoning, it becomes clear that suggesting disinfectants as a COVID-19 treatment is not only scientifically inaccurate but also extremely harmful to the human body. Current detection models often lack specialized domain knowledge and the capacity for such reasoning, limiting their ability to provide meaningfully and contextually accurate explanations for their predictions.

To address these challenges, we draw inspiration from the process of writing debunking articles, which involves gradually uncovering the truth by understanding the content, gathering relevant domain information, and combining these insights to debunk the rumor effectively. To this end, we present **ExMRD**, a novel **Ex**plainable **M**icro-video **R**umor **D**etection framework, which can provide clear and well-reasoned explanations for MVRD. Specifically, we introduce $R^3$CoT, a novel Chain-of-Thought (CoT) [52] inference mechanism. $R^3$CoT consists of three key steps: *Refining*, *Retrieving*, and *Reasoning*. Specifically, at the refining step, the Multimodal Large Language Model (MLLM) is prompted to reorganize low-quality and misleading video content from both textual and visual perspectives, producing a coherent representation of the content in the video. At the retrieving step, rumor-related domain knowledge is generated based on the refined content, enriching the video's context for rumor detection. At the final reasoning step, logical inference is applied by cross-verifying the refined content with domain knowledge, providing evidence to support or refute the video's authenticity. Fig. 1 demonstrates the outputs produced after each step of the $R^3$CoT mechanism, specifically showing the refined content, the retrieved domain knowledge, and the reasoning behind the final conclusion.

While fine-tuning MLLMs with $R^3$CoT guidance can greatly enhance their performance in MVRD, this fine-tuning process introduces significant computational overhead, which significantly limits its practicality in real-world applications. To address this, we introduce a novel *Small Language Reviewer* (SLReviewer), which acts as a reliable "reviewer" within ExMRD. The main idea is to refine MLLM outputs using distilled knowledge from the proposed $R^3$CoT mechanism, ensuring reliable predictions with low computational overhead. By fine-tuning SLMs, rather than MLLMs, we achieve a balance between performance and efficiency, making ExMRD more practical for deployment.

Our main contributions are summarized as follows:

- **An explainable MVRD framework** that generates explicit rationales behind rumor detection. This work is the first to incorporate explainability into MVRD by designing the reasoning step of $R^3$CoT, which produces clear rationales for determining whether a video contains rumor content. This enhances transparency and interpretability, making the decision-making process in MVRD more accessible to users and moderators.

- **A novel $R^3$CoT mechanism** that enables MLLMs to perform refining, retrieving, and reasoning for explainable conclusions. This mechanism helps address rumor detection errors caused by inconsistent video quality, and lack of domain knowledge and reasoning, providing a comprehensive understanding of video content in MVRD.

- **An efficient SLReviewer** that improves both efficiency and prediction reliability while balancing performance with computational resources. SLReviewer is fine-tuned using insightful distilled knowledge from the MLLM, enabling it to generate credible and accurate predictions.

Extensive experiments on real-world micro-video datasets demonstrate that our ExMRD outperforms state-of-the-art baselines while providing clear and well-reasoned rationales. Notably, our ExMRD achieves an average improvement of 5.37% in Macro F1 across all three datasets, outperforming 13 competitive baselines. The code and data to reproduce the results are available at https://anonymous.4open.science/r/ExMRD and will be made public later.

## 2 Related Work

### 2.1 Micro-Video Rumor Detection

The task of MVRD focuses on identifying rumor content by analyzing multiple modalities within micro-videos, such as text, vision, and audio. Early detection methods primarily relied on unimodal information [24, 35, 41]. For example, Papadopoulou et al. [35] utilized basic video metadata and user engagement features, while Serrano et al. [41] analyzed user comments, focusing on conspiracy-related remarks as key indicators. The complexity and richness of micro-video content – where various modalities often interplay – make unimodal approaches inadequate for accurate rumor detection. Recently, FakeSV [36] improves content representation by leveraging cross-modal correlations and integrating social context. Despite these advancements, current methods in MVRD remain black-box models, providing only the final detection result without offering any interpretability for the decision-making process, which increases a lot of risks and limits user trust. In contrast, our study aims to provide clearer explanations and enhance transparency for both moderators and viewers, addressing the critical need for explainability in micro-video rumor detection. – i.e., understand, trust, and act upon the detection outcomes.

### 2.2 Chain-of-Thought Prompting

CoT prompting [52] has been developed to guide Large Language Models (LLMs), such as GPT-3 [7], LLaMA [46], to better understand tasks and generate better responses by encouraging step-by-step reasoning. Moreover, the recent OpenAI-o1 [56] has demonstrated remarkable reasoning capabilities by deeply integrating CoT into its architecture, enabling the model to seamlessly perform multi-step reasoning. Specifically, various CoT strategies [30, 51, 57] have been introduced to enhance the reasoning abilities of LLMs. With the advancement of MLLM [26, 34], CoT prompting has been adapted to improve reasoning across both visual and textual modalities [10, 31, 40]. This adaptation enhances MLLM's ability to synthesize information from multiple modalities and align their representations, leading to improved performance on multimodal tasks.

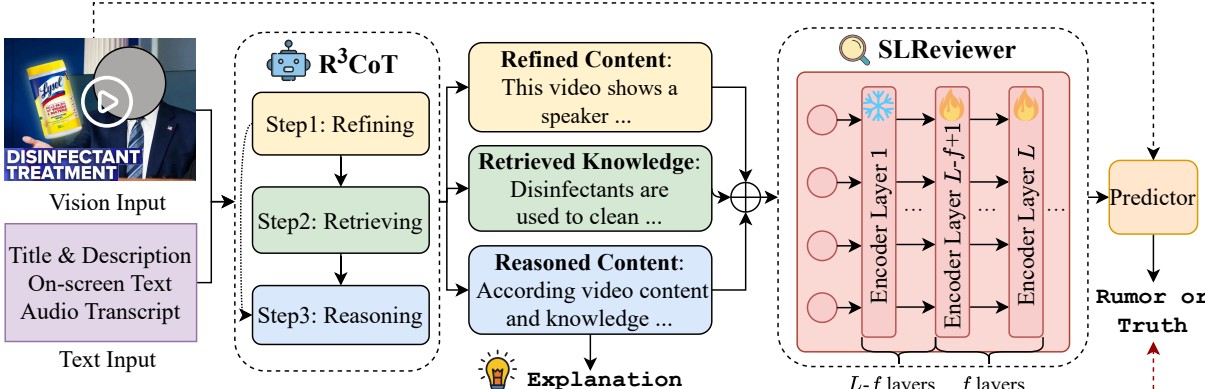

Fig. 2: The structure of our proposed ExMRD framework. (1) The R³CoT process prompts MLLMs to refine the video content, retrieve domain knowledge, and reason to provide explanations. (2) The SLReviewer is to distill the explainable evidence from R³CoT to facilitate reliable rumor detection.

However, existing works primarily leverage CoT for generating better responses in scenarios like visual question answering [29, 43, 53] and mathematical reasoning [16, 19, 45], overlooking the potential of CoT to enhance task-specific explainability. In contrast, our ExMRD incorporates a CoT based inference mechanism, R³CoT, to guide MLLMs to generate accurate, understandable, and interpretable predictions via the designed three key steps: *Refining*, *Retrieving*, and *Reasoning*. To our knowledge, this work is among the first to leverage the reasoning capabilities of CoT to prompt MLLMs in delivering precise and interpretable predictions for MVRD.

## 3 Methodology

**Problem Definition**. Let $\mathcal{M}$ represent a micro-video on video platforms. The video $\mathcal{M}$ is characterized by its multimodal content, which includes textual, visual, and audio modalities, denoted as $\mathcal{M} = \{\mathcal{T}, \mathcal{V}, \mathcal{A}\}$. The primary objective of MVRD is to determine whether the video $\mathcal{M}$ contains the *rumor* content by simultaneously considering all its modalities $\mathcal{T}$, $\mathcal{V}$, and $\mathcal{A}$.

**Overview**. First, we design the R³CoT, a three-step CoT inference mechanism: (1) *Refining*. Prompt the MLLMs to reorganize chaotic and misleading video content into a well-structured format, serving as the foundation for subsequent generation; (2) *Retrieving*. Instruct the MLLMs to generate relevant domain knowledge based on the refined video content to support the final reasoning; (3) *Reasoning*. Guide the MLLMs to apply logical inference to cross-verify the refined content with domain knowledge, providing evidence to confirm or refute the authenticity of the micro-video. Second, we introduce the SLReviewer, to distill the reasoning output from the MLLMs, ensuring reliable final predictions. An overview of our proposed ExMRD is shown in Fig. 2.

## 3.1 R³CoT Mechanism

*3.1.1 Feature extraction.* For the textual modality $\mathcal{T}$, we extract the video's metadata (title and description) $\mathcal{T}_m$ and the on-screen text $\mathcal{T}_o$. For visual modality $\mathcal{V}$, we uniformly sample $k$ frames from each video: $\mathcal{V}_f = \{v_1, \ldots, v_k\}$. For audio modality $\mathcal{A}$, we convert

the audio into transcript denoted as $\mathcal{A}_t$. The detailed process of feature extraction is summarized in Appendix D.3.

*3.1.2 Refining.* In the domain of rumor video detection, a diverse range of micro-video creators exists, spanning from professional media outlets to ordinary individual users, leading to micro-videos of varying quality. Micro-videos frequently contain chaotic and misleading content, making detection challenging, thus requiring the reorganization of the presented information. To address these problems, the initial step in the R³CoT mechanism involves refining the video content to generate well-structured representations by reorganizing both textual and visual elements for clearer analysis.

First, from the textual perspective, we combine the metadata text $\mathcal{T}_m$, on-screen text $\mathcal{T}_o$, and audio transcript $\mathcal{A}_t$ as the textual content of the video. However, due to elaborate video effects, diverse typographic styles, and non-news elements (e.g., watermarks, creator attributions, and platform identifiers), the on-screen text often contains numerous recognition errors. To mitigate this issue, we empower the MLLM $\mathcal{F}(\cdot)$ to focus on rumor elements while enhancing on-screen accuracy through in-context learning. We restore the original rumor content from a textual perspective, resulting in refined textual content $\mathcal{R}_\text{text}$. This step can be formulated as follows:

$$\mathcal{R}_\text{text} = \mathcal{F}([\mathcal{T}_m; \mathcal{T}_o; \mathcal{A}_t], \text{Prompt}_1), \tag{1}$$

where $[;]$ denotes the concatenation operation, with the template $\text{Prompt}_1$ described in Step 1 of Fig. 3.

Second, from the visual perspective, we develop a visual-centric strategy that focuses on the refining of scene content in the micro-video. This strategy aims to filter out subjective elements, such as subtitles or auditory narratives, focusing solely on the visual information presented in the videos. By prioritizing these visual aspects, we seek to provide a more objective and comprehensive understanding of the events depicted, offering insights that may not be explicitly conveyed in the textual content.

Instead of feeding individual frames into the MLLM, we propose to construct $n$ composite frames $\mathcal{P}_v = \{P_1, P_2, \cdots, P_n\}$. Each composite frame $P_i$ consists of an $m \times m$ grid of consecutive frames from

**Step1 Refining**

**Prompt₁** (Textual Perspective): Analyze {Text Input: disinfectant treatment to kill virus...} to reconstruct video content, correcting errors and enhancing coherence while preserving key details.

**Answer:** This video shows a speaker suggesting ... using disinfectants internally, such as through injection...

**Prompt₂** (Visual Perspective): Analyze {Vision Input: composite frames of video} to generate a descriptive caption, focusing solely on key visual elements and events while ignoring any on-scree-text and subjective elements.

**Answer:** This video content shows a public figure is giving a speech, discussing issues related to disinfectants...

**Step2 Retrieving**

**Prompt₃**: Analyze the {Refined Content: This video shows ...} from both textual and visual descriptions of the micro-video. Use your pre-trained knowledge to provide relevant background information, enhancing comprehension of the video context, without assessing authenticity. Ensure responses are concise, and focused.

**Answer:** The knowledge related including: disinfectants are used to clean surfaces ...

**Step3 Reasoning**

**Prompt₄**: Analyze the {Refined Content: This video shows ...}, and {Relevant Knowledge: disinfectants are...} from a multimodal perspective. Systematically deconstruct the video's structure and argumentation, identifying any logical flaws or weak points. Focus on elucidating the logical framework without assessing veracity.

**Answer:** According refined content and related knowledge, injecting disinfectants to kill virus ... harm to the body

Fig. 3: An illustration of three-step prompting of our proposed R³CoT. Step 1, prompt MLLM to organize chaotic and misleading video input; Step 2, instruct MLLM to generate rumor-relevant knowledge; Step 3, guide MLLM to conduct inference to cross-verify the refined and retrieved content.

the initial video frames $\mathcal{V}_f$: $P_i = [v_{i_1}, v_{i_2}, \ldots, v_{i_{m \times m}}]$. This allows the model to better grasp temporal changes and scene dynamics. The generation process of refined visual content $\mathcal{R}_{\text{vision}}$ is guided by the template prompt₂ presented in Step 1 of Fig. 3,

$$\mathcal{R}_{\text{vision}} = \mathcal{F}(\mathcal{P}_v, \text{Prompt}_2). \quad (2)$$

The refined textual and visual content will be used as the input information for the next step to generate the domain knowledge for the micro-video.

*3.1.3 Retrieving.* This step aims to instruct the MLLM to generate expressive and relevant domain knowledge for the given micro-video based on the refined content. The retrieved domain knowledge enables the MLLM to better understand the rumors presented in the micro-video, thus facilitating effective reasoning. Specifically, let $\mathcal{R}_{\text{refining}} = [\mathcal{R}_{\text{text}}; \mathcal{R}_{\text{vision}}]$ represent the refined video content. We guide the domain knowledge retrieval process using the template Prompt₃ in Step 2 of Fig. 3. This step can be formulated as follows:

$$\mathcal{R}_{\text{retrieving}} = \mathcal{F}(\mathcal{R}_{\text{refining}}, \text{Prompt}_3). \quad (3)$$

The expressive domain knowledge and the refined content will jointly help the MLLM to make the final reasoning in the next step.

*3.1.4 Reasoning.* This step utilizes logical inference to cross-verify the refined content with the domain knowledge from the previous steps, aiming to uncover evidence that either confirms or refutes the authenticity of the video. Furthermore, this process enhances the explainability of the model by providing clear evidence (e.g., conflicting or consistent facts) to explain why content is classified as a rumor, thereby providing transparency into the model's decisions. We guide the logical reasoning using the template Prompt₄ in Step 3 of Fig. 3. This step can be formulated as follows:

$$\mathcal{R}_{\text{reasoning}} = \mathcal{F}([\mathcal{R}_{\text{refining}}; \mathcal{R}_{\text{retrieving}}], \text{Prompt}_4). \quad (4)$$

By leveraging the logically rigorous and coherent R³CoT mechanism, our framework **addresses the limitations** of prior works [8, 9, 36, 42], which typically overlook the chaotic and low-quality nature of video content and lack the necessary rumor-specific background knowledge and deep reasoning required for accurate predictions. **In contrast**, our R³CoT mechanism equips ExMRD with deeper insights through refining the raw video content, retrieving the domain knowledge, and reasoning the deep rationales, ultimately achieving notable improvements in prediction accuracy.

## 3.2 Small Language Reviewer

Although the MLLM can make predictions through our carefully designed three-step CoT process, its inference remains unreliable due to the inherent limitations of large language models [5, 18], as the predictions may suffer from hallucinations and not be faithful to the video content and reasoning process. While fine-tuning the MLLM for MVRD could help mitigate this issue, it is not practical due to the vast number of parameters of the MLLM and the associated huge computational cost.

To this end, we further propose the Small Language Reviewer (SLReviewer), which distills the outputs from the MLLM into a smaller language model. By fine-tuning the smaller language model, a more reliable and practical solution is achieved, as it requires significantly fewer parameters and is computationally efficient. The fine-tuning process of SLM can be represented as:

$$\mathcal{S}(\mathbf{x}) = \mathcal{L}_L \circ \mathcal{L}_{L-1} \circ \cdots \circ \mathcal{L}_{L-f+1} \circ \mathcal{L}_{L-f}^{\text{fixed}} \circ \cdots \circ \mathcal{L}_1^{\text{fixed}}(\mathbf{x}), \quad (5)$$

where $\mathcal{L}_i$ represents the $i$-th transformer encoder layer. The parameters of the last $f$ layers, denoted as $\mathcal{L}_{L-f+1}$ to $\mathcal{L}_L$, are fine-tuned during training. Meanwhile, the remaining layers are frozen, enabling the SLReviewer to retain its ability to review the output produced by the MLLM. The refined content $\mathcal{R}_{\text{refining}}$, domain

knowledge $\mathcal{R}_{\text{retrieving}}$, and reasoning $\mathcal{R}_{\text{reasoning}}$ are concatenated and fed into the SLReviewer to generate the textual feature representation $\mathbf{H}_t \in \mathbb{R}^{l \times d_t}$ for the final prediction. This process can be written as:

$$\mathbf{H}_t = \mathcal{S}[\mathcal{R}_{\text{refining}}; \mathcal{R}_{\text{retrieving}}; \mathcal{R}_{\text{reasoning}}]. \tag{6}$$

We employ the pre-trained Vision Transformer (ViT) [14] to extract visual features $\mathbf{H}_v \in \mathbb{R}^{k \times d_v}$ from the video frames $\mathcal{V}_f$. To align the sequence lengths of the textual features $\mathbf{H}_t$ and the visual features $\mathbf{H}_v$, we apply an average pooling strategy, resulting in $\bar{\mathbf{H}}_t \in \mathbb{R}^{d_t}$ and $\bar{\mathbf{H}}_v \in \mathbb{R}^{d_v}$. Subsequently, a two-layer Multilayer Perceptron (MLP) is used as a predictor to fuse the pooled textual and visual features, producing the final prediction: $\hat{y} = \text{Predictor}[\Psi_t(\bar{\mathbf{H}}_t); \Psi_v(\bar{\mathbf{H}}_v)]$, where $\Psi_t(\cdot)$ and $\Psi_v(\cdot)$ denote the linear mapping functions.

For model training, we only need to optimize the parameters of SLReviewer. Specifically, we adopt Binary Cross-Entropy loss to optimize the model. In addition, the efficiency analysis, training algorithm, and mathematical proof of ExMRD, are provided in Appendix A-C.

**Table 1: Statistics of three datasets.**

| Dataset | Time Range | # Rumor | # Truth | # Total | Duration (s) |
|---------|-----------|---------|---------|---------|--------------|
| FakeSV | 2017/10-2022/02 | 1,810 | 1,814 | 3,624 | 39.88 |
| FakeTT | 2019/05-2024/03 | 1,172 | 819 | 1,991 | 47.69 |
| FVC | 2016/01-2018/01 | 1,633 | 1,131 | 2,764 | 87.83 |

## 4 Experiments

An overview of the experimental setup is outlined below, with detailed descriptions of the datasets, baseline models, and implementation available in Appendix D.

- **Datasets**. To analyze the effectiveness of our ExMRD, we conduct experiments on three real-world micro-video datasets: FakeSV [36], FakeTT [8], and FVC [35]. Table 1 summarizes the detailed statistics of three datasets. Following existing works [8, 36], we employ a temporal split strategy to simulate real-world scenarios on micro-video platforms. In this strategy, we divide each dataset chronologically into training, validation, and test sets, with respective ratios of 70%, 15%, and 15%.

- **Baselines**. To verify the superiority of ExMRD, we compare it against 13 competitive baselines, which can be categorized into three groups: (1) *Unimodal detection methods* which utilize single modality (e.g., textual modality) of micro-videos to conduct detection: BERT [13], ViT [14], MFCC [12] and TSformer [6]; (2) *Multimodal detection methods* which incorporate all modalities to improve the precision in detecting rumors in micro-videos: TikTec [42], FANVM [11], CAFE [9], HMCAN [38], SV-FEND [36] and FakingRec [8]; (3) *MLLM based methods* which employ the latest released advanced MLLMs to detect the video rumor: GPT-4o-m [34], LLaVA-OV [22], Qwen2-VL [50].

- **Model Implementation**. We employ GPT-4o-m [34] as the MLLM backbone because it is scalable and easy to deploy. Additionally, BERT [13] is adopted as the SLM backbone due to its robust contextual understanding and performance in many natural language processing tasks.

- **Metrics**. Following prior works [8, 36], we employ four metrics to evaluate the performance: Accuracy (**ACC**), Macro F1 score (**M-F1**), Macro Precision (**M-P**), and Macro Recall (**M-R**).

### 4.1 Overall Performance

To assess the effectiveness of our ExMRD, we compare ExMRD with 13 competitive baselines. The results are summarized in Table 2. From the results, we have the following observations.

First, ExMRD consistently outperforms all baselines across various metrics on three datasets, showing an average improvement of 4.99% in Accuracy and 5.37% in Macro F1. To further verify its effectiveness, ExMRD and the strongest baseline are retrained five times, with the resulting $p$-values, all below 0.05, confirming the statistical significance of ExMRD's improvement. These gains are attributed to ExMRD's innovative $R^3$CoT mechanism, which refines content, retrieves domain knowledge, and applies reasoning. Furthermore, SLReviewer efficiently utilizes the distilled knowledge from MLLMs, yielding precise predictions with minimal computational overhead.

Second, the unimodal methods show significantly lower performance, highlighting the importance of multimodal information in MVRD. Among them, BERT, which leverages the textual information for prediction, outperforms other methods in this group. This indicates that textual modality contains more semantic information and is more conducive to rumor detection. Our ExMRD framework leverages extensive textual data from multiple sources (title, description, on-screen text, and audio transcript), with the $R^3$CoT refining step enhancing the text quality for improved predictions.

Third, multimodal methods generally outperform unimodal approaches, demonstrating the benefit of combining text and visuals in MVRD. FakingRec, for example, achieves strong results by focusing on the content creation process, leading to a robust multimodal understanding. However, ExMRD surpasses all multimodal baselines by refining video content to tackle the low-quality inputs. Additionally, accurately classifying a video as rumor or truth requires domain knowledge and logical reasoning, aspects that other models overlook.

Fourth, MLLM-based methods excel in zero-shot multimodal tasks but their lack of task-specific adaptation often leads to inconsistent performance in MVRD, making them insufficient for the detection. Direct fine-tuning MLLMs is computationally expensive, limiting practicality. In our ExMRD, we propose SLReviewer to refine MLLM outputs, providing more reliable predictions. Consequently, ExMRD achieves superior performance in MVRD compared to conventional MLLM-based methods.

### 4.2 Ablation Study

We conduct experiments to explore the impact of each main component in ExMRD, with the results summarized in Table 3.

*4.2.1 Effect of $R^3$CoT Mechanism.* To assess the impact of the $R^3$CoT mechanism, each step of $R^3$CoT is removed individually to evaluate its effect. Specifically, the following ablation studies are conducted: (1) **w/o Refine**, where the refined output is replaced with the original video content; (2) **w/o Retrieve**, where the domain knowledge is excluded; (3) **w/o Reason**, where the reasoning output is removed; and (4) **w/o $R^3$CoT**, where the distilled knowledge from MLLM is replaced as the original video content. The results indicate that each steps play pivotal roles in detecting rumor in micro-videos and provide insightful rationales with the prediction result. Moreover, a substantial performance drop is observed when the entire $R^3$CoT mechanism is removed, further validating

**Table 2: Performance comparison on three real-world datasets. The best results are in highlighted red bold, while the second results are in black bold. Higher values of Accuracy, Macro F1, Macro Precision, and Macro Recall signify better performance.**

| Dataset | FakeSV | | | | FakeTT | | | | FVC | | | |
|---|---|---|---|---|---|---|---|---|---|---|---|---|
| Model | ACC | M-F1 | M-P | M-R | ACC | M-F1 | M-P | M-R | ACC | M-F1 | M-P | M-R |
| BERT | 80.63 | 80.14 | 80.56 | 79.90 | 71.24 | 69.31 | 68.98 | 70.85 | 69.29 | 68.13 | 68.72 | 67.95 |
| ViT | 71.22 | 71.04 | 71.04 | 71.33 | 65.55 | 64.39 | 65.17 | 67.11 | 81.54 | 80.74 | 82.03 | 80.25 |
| MFCC | 61.07 | 61.05 | 61.64 | 61.74 | 52.51 | 52.23 | 64.26 | 62.21 | 65.05 | 60.79 | 65.75 | 61.68 |
| TSformer | 72.14 | 71.95 | 71.91 | 72.20 | 64.88 | 64.69 | 68.79 | 70.43 | 90.92 | 90.67 | 91.06 | 90.40 |
| TikTec | 73.43 | 73.26 | 73.23 | 73.54 | 66.22 | 65.08 | 65.84 | 67.87 | 74.60 | 74.54 | 74.63 | 74.52 |
| FANVM | 79.52 | 78.81 | 79.81 | 78.46 | 71.57 | 70.21 | 70.22 | 72.63 | 79.27 | 77.41 | 82.47 | 76.77 |
| CAFE | 71.03 | 71.00 | 71.41 | 71.67 | 69.57 | 67.91 | 67.83 | 69.85 | 83.59 | 83.12 | 83.76 | 82.79 |
| HMCAN | 79.52 | 78.81 | 79.81 | 78.46 | 68.56 | 68.41 | 72.78 | 74.72 | 85.62 | 84.99 | 86.48 | 84.40 |
| SV-FEND | 80.88 | 80.54 | 80.18 | 80.62 | 77.14 | 75.63 | 75.12 | 77.56 | 87.59 | 87.36 | 87.34 | 87.40 |
| FakingRec | **84.69** | **84.30** | **84.80** | **84.01** | **79.60** | **77.76** | **77.12** | **78.88** | 90.92 | 90.78 | 90.65 | 90.95 |
| GPT-4o-m | 66.42 | 65.88 | 65.90 | 65.87 | 57.85 | 57.78 | 62.91 | 63.65 | 66.11 | 65.51 | 65.49 | 65.54 |
| LLaVA-OV | 57.54 | 50.71 | 61.57 | 55.94 | 46.82 | 46.81 | 53.44 | 53.36 | 60.21 | 56.55 | 58.91 | 57.30 |
| Qwen2-VL | 53.85 | 53.72 | 54.29 | 54.20 | 53.18 | 52.95 | 56.77 | 57.35 | 59.15 | 58.05 | 58.15 | 58.02 |
| **ExMRD** | **86.90** | **86.52** | **87.31** | **86.13** | **84.28** | **83.13** | **82.27** | **85.19** | **96.82** | **96.75** | **97.02** | **96.75** |
| Improv. | 2.61%↑ | 2.63%↑ | 2.96%↑ | 2.52%↑ | 5.88%↑ | 6.91%↑ | 6.68%↑ | 8.00%↑ | 6.49%↑ | 6.58%↑ | 7.03%↑ | 6.38%↑ |
| $p$-val. | $2.26e^{-3}$ | $2.33e^{-3}$ | $3.17e^{-3}$ | $2.42e^{-3}$ | $1.64e^{-2}$ | $9.36e^{-3}$ | $5.57e^{-3}$ | $2.82e^{-3}$ | $4.94e^{-4}$ | $5.62e^{-4}$ | $3.18e^{-4}$ | $1.07e^{-3}$ |

**Table 3: Ablation study on key components of ExMRD.**

| Dataset | FakeSV | | FakeTT | | FVC | |
|---|---|---|---|---|---|---|
| Variant | ACC | M-F1 | ACC | M-F1 | ACC | M-F1 |
| w/o Refine | 82.10 | 81,81 | 83.28 | 82.22 | 94.25 | 94.90 |
| w/o Retrieve | 85.05 | 84.60 | 81.61 | 80.47 | 93.34 | 93.11 |
| w/o Reason | 85.05 | 84.42 | 79.60 | 78.62 | 95.31 | 95.20 |
| w/o $R^3$CoT | 80.07 | 79.72 | 80.60 | 79.58 | 93.65 | 93.47 |
| w/o Fine-tune | 84.87 | 84.27 | 81.61 | 80.47 | 92.59 | 92.33 |
| MLP-based | 85.42 | 84.86 | 78.93 | 77.63 | 95.31 | 95.20 |
| **ExMRD** | **86.90** | **86.52** | **84.28** | **83.13** | **96.82** | **96.75** |

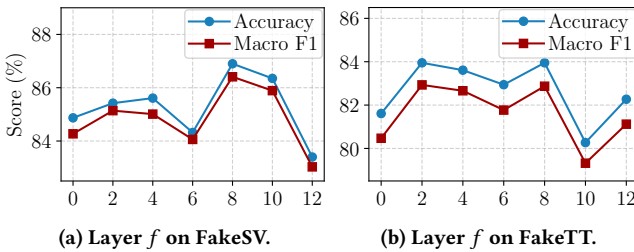

(a) Layer $f$ on FakeSV.  (b) Layer $f$ on FakeTT.

**Fig. 4: Sensitivity analysis of the number of fine-tuning decoder layers $f$ on the FakeSV and FakeTT datasets.**

the critical role of the distilled knowledge obtained from $R^3$CoT guided MLLM in MVRD.

*4.2.2 Effect of SLReviewer.* To assess the effectiveness of the SLReviewer, two ablation variants are developed: (1) **w/o Fine-tune**, where no fine-tuning is applied to the SLM; (2) **MLP-based Reviewer**, where a one-layer MLP is attached to the last layer of the frozen SLM. The results indicate the necessity of fine-tuning SLReviewer to adapt to the distilled outputs produced by the MLLM, as freezing all the layers leads to significantly degraded performance. Moreover, the MLP-based reviewer struggles to capture the intricate reasoning patterns required for this task, leading to a substantial drop in performance. This is likely because the shallow architecture of the MLP lacks the representational capacity needed to model the nuanced interactions between the layers of the pre-trained SLM.

## 4.3 Hyper-Parameter Sensitivity Analysis

In this section, sensitivity analysis of hyper-parameters within ExMRD is conducted on the FakeSV and FakeTT datasets, with the results presented in Fig. 4. The results show that as the number of fine-tuning decoder layers in SLReviewer increases, the performance of ExMRD improves initially, demonstrating that SLReviewer can effectively self-update its parameters to adapt to the

reasoning patterns derived from $R^3$CoT and generate more accurate outputs. However, fine-tuning too many layers can disrupt the rich knowledge learned during SLM pre-training, leading to a decline in performance. To balance the preservation of pre-trained knowledge with adaptation to newly distilled knowledge, the number of fine-tuning decoder layers $f$ is set to 8 for both datasets. Additional parameter analysis is provided in Appendix E.3.

## 4.4 Model Generalizability Analysis

In this section, we explore the generalizability of ExMRD from two distinct perspectives. First, we evaluate the generalizability of the main components within ExMRD– specifically the $R^3$CoT and SLReviewer– to determine their effectiveness across different MLLMs. Subsequently, we investigate the model's generalizability from a dataset perspective, focusing on its ability to train on one dataset and consistently perform well on other distinct datasets.

*4.4.1 Generalizability of Main Components.* We evaluate the generalizability of our main components, $R^3$CoT and SLReviewer, across various MLLMs for MVRD. Results for the FakeSV and FakeTT datasets are shown in Fig. 5, while the results for FVC, due to page limitations, are presented in Fig. 10. As observed, integrating $R^3$CoT significantly enhances the accuracy of various MLLMs in distinguishing rumors from truths in micro-videos, underscoring

 

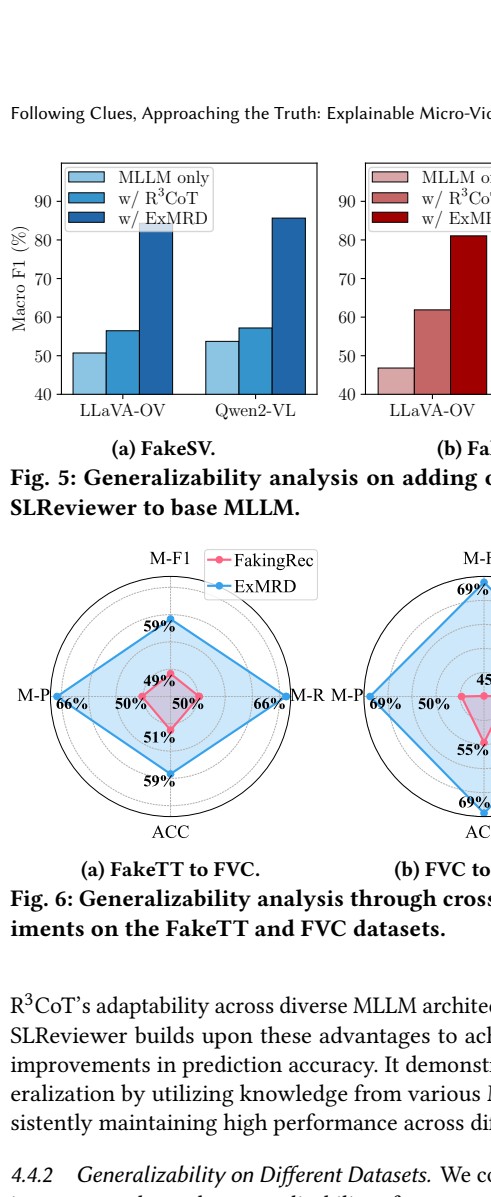

(a) FakeSV.  (b) FakeTT.

Fig. 5: Generalizability analysis on adding our $R^3CoT$ and SLReviewer to base MLLM.

Fig. 6: Generalizability analysis through cross-dataset experiments on the FakeTT and FVC datasets.

(a) FakeTT to FVC.  (b) FVC to FakeTT.

$R^3CoT$'s adaptability across diverse MLLM architectures. Moreover, SLReviewer builds upon these advantages to achieve significant improvements in prediction accuracy. It demonstrates strong generalization by utilizing knowledge from various MLLMs and consistently maintaining high performance across different datasets.

*4.4.2 Generalizability on Different Datasets.* We conduct the experiments to evaluate the generalizability of our proposed ExMRD and the most competitive baseline model FakingRec, and the results are reported in Fig. 6. In this experiment, we select the FakeTT and FVC datasets, which are sourced from different platforms and varying significantly in content style and target audience, to conduct cross-dataset evaluations. Specifically, we train and validate the models on one dataset and test them on the other. The results show that our ExMRD strongly outperforms FakingRec across all metrics in both cross-dataset evaluations. The baseline model struggles to handle dataset biases, such as superficial multimodal features and platform-specific characteristics. In contrast, our model effectively bridges the gap between inconsistencies in micro-video quality through the MLLM by the guidance of the refining step in $R^3CoT$. Moreover, the MLLM is instructed by the retrieving and reasoning steps from $R^3CoT$ to generate the domain knowledge and the rationale for detecting rumors for the micro-videos in the target dataset. This distilled knowledge demonstrates significant generalizability and is fed to SLReviewer to make the precise prediction. These observations further confirm the superiority of ExMRD in handling diverse videos with varying quality and its robustness in cross-platform scenarios, making it well-suited for real-world deployment.

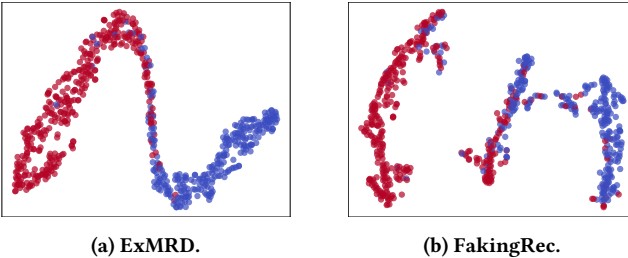

(a) ExMRD.  (b) FakingRec.

Fig. 7: t-SNE visualization of ExMRD and FakingRec on the FVC dataset. Red points represent rumors; blue points, truth.

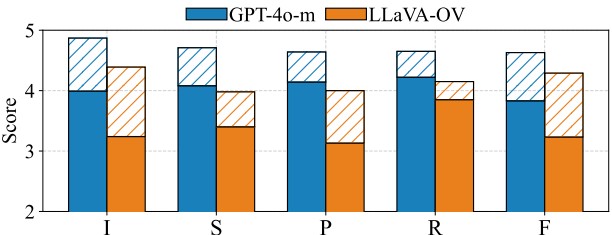

Fig. 8: Comparison of explanation quality with and without proposed $R^3CoT$ on the FakeTT dataset. I: Informativeness, S: Soundness, P: Persuasiveness, R: Readability, F: Fluency.

## 4.5 Model Prediction Visualization

Fig. 7 visualizes the embedding distribution of the two categories (i.e., Rumor and Truth) on the test set of the FVC dataset, using t-SNE [47]. In this study, we select the output from the last layer of the classifier in our model as the embedding. We observe that our ExMRD produces more discriminative representations, with clearer boundaries between instances of different labels. This result underscores ExMRD's ability to generate the evidence of whether the video is rumor or truth through the $R^3CoT$ and distill this evidence to the SLReviewer to perform accurate predictions.

## 4.6 Model Explainability

To assess the explainability of ExMRD, we first compare the quality of explanations generated by various MLLMs, with and without $R^3CoT$. A case study on selected micro-videos then demonstrates how effectively ExMRD explains its classifications.

*4.6.1 Quality of Explainability.* In this section, we assess the contribution of the proposed $R^3CoT$ to the quality of explanation (i.e., reasoning output). We employ G-Eval [27], an LLM-based reference-free evaluation approach, to evaluate the text quality of the explanations generated by our framework by comparing the base MLLM with or without $R^3CoT$. To be specific, we utilize the following criteria [48, 49]: (1) Informativeness: the explanation provides new information, such as explaining the background and additional context; (2) Soundness: the explanation seems valid and logical; (3) Persuasiveness: the explanation is convincing; (4) Readability: the explanation follows proper grammar and structural rules; (5) Fluency: the explanation flows smoothly with coherent and well-connected ideas. For each criterion, a 5-point Likert scale [20] is employed, where 1 meant the poorest quality and 5 the best.

Fig. 8 illustrates the average improvement in explanation quality, as evaluated by G-Eval, for base MLLM with and without $R^3CoT$, across five criteria. The results show that: (1) in *informativeness* and

**Table 4: Case study of correctly detected micro-video rumors on the FakeTT dataset.**

| | Case 1 | Case 2 |
|---|---|---|
| **Micro-video** | 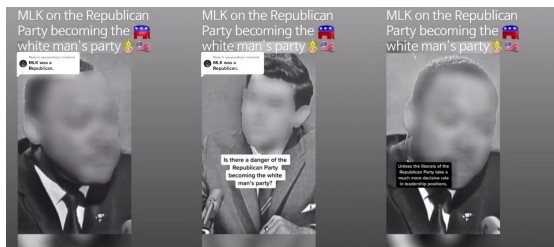 | |
| **Viewpoint** | Three Simple Ways to Check Food Quality | Martin Luther King was not a Republican |
| **Original Content** | REAL FOOD VS FAKE FOOD CHECK HOW NATURAL PRODUCTS AREI ARTIFICIAL REAL HONEY HONEY NATURAL ... | Replying to marywesling Dr. King did not associate himself as a member of any party. #mlk #mlkday |
| **Refining** | This video presents a comparison between real and artificial food products. It ... identify natural products versus artificial alternatives, with examples like real honey versus ... | The micro-video shows Dr. Martin Luther King Jr. expressed concern about the Republican Party potentially becoming ..." ... He acknowledged that this trend poses a significant danger ... |
| **Retrieving** | Food Authenticity Checks: Common methods include testing for natural chemical markers (e.g., pure honey vs. adulterated), and observing physical characteristics during cooking... | Dr. Martin Luther King Jr. was a renowned civil rights leader... he did not publicly declare himself a member of any political party... His main focus ...not partisan politics ... |
| **Reasoning** | The argument seems visually driven ... Cooking appearance alone may not conclusively differentiate between natural and artificial, as processed foods can mimic the appearance of natural ones. | From the video, Dr. Jin did express concerns during an interview about the Republican Party potentially becoming a "white party." This is consistent with historical records ... |
| **Ground Truth** | Rumor | Truth |
| **ExMRD** | Rumor ✓ | Truth ✓ |

soundness, MLLM equipped with R³CoT exhibit significant improvement over the original MLLM, underscoring the necessity of R³CoT for providing expressive domain knowledge during the retrieving step; (2) in *readability* and *fluency*, MLLM equipped with R³CoT outperform the original versions, demonstrating the effectiveness of refining step in reorganizing video content and enhancing clarity; (3) in *persuasiveness*, the MLLM with R³CoT displays a significant improvement, suggesting that it contributes to more convincing rationales through its reasoning step.

*4.6.2 Qualitative Analysis on Explainability.* To further investigate the explainability of our proposed ExMRD, we randomly selected two micro-video cases from FakeTT to explore how ExMRD classifies each video as either a rumor or truth, as shown in Table 4.

In case 1, a rumor micro-video claims that *Three Simple Ways to Check Food Quality*. The original content involves cooking two types of food in a frying pan to test whether they are natural products. ExMRD first refines this content to highlight the core claim. Subsequently, it retrieves domain knowledge on common methods for verifying food authenticity, such as testing for natural chemicals, and applies logical reasoning to reveal that the content primarily relies on visual appeal to attract viewers, without offering valid techniques for determining whether the food is genuinely natural. As a result, our ExMRD correctly classifies the content in this video as a *Rumor*. This case demonstrates how ExMRD not only detects misinformation but also offers a well-founded rationale supported by factual knowledge and logical analysis.

In Case 2, a real micro-video refutes the statement that *Martin Luther King was a Republican*. The original content includes a description of Dr. King's concerns that the Republican Party might become a *white party*. Our framework refines the text to emphasize that Dr. King expressed concerns about the party's direction but did not publicly align himself with any political party. It retrieves relevant domain knowledge confirming that Dr. King was a non-partisan civil rights leader, and uses logical reasoning to align this information with historical records. Consequently, the framework accurately classifies the video as *Truth*. This case highlights how ExMRD integrates historical context and logical reasoning to verify the authenticity of the claim. Additional qualitative analyses and error analysis are provided in Appendix E.5-F.

## 5 Conclusion

This work introduces ExMRD, an Explainable framework for interpretable Micro-video Rumor Detection. The proposed R³CoT mechanism in ExMRD is a novel three-step CoT process – Refining, Retrieving, and Reasoning – that reorganizes the raw video content, retrieves rumor-related domain knowledge, and generates explainable conclusions on whether the micro-video contains misleading information. Instead of directly fine-tuning MLLMs, which is computationally expensive, we propose SLReviewer within the ExMRD framework, distilling CoT-guided MLLM outputs to ensure accurate predictions with minimal computational overhead, making it more adaptable to real-world demands. Extensive experiments conducted on three real-world datasets demonstrate the effectiveness of ExMRD in both rumor detection and explainability. We believe that ExMRD is a valuable tool for detecting rumors on various micro-video platforms (e.g., TikTok and YouTube Shorts) while also promoting AI transparency and fostering a more trustworthy and safer online experience.

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

## A   Efficiency Analysis

In this section, we provide a comparison of performance on macro F1 with regard to the number of trainable parameters for ExMRD and the other competitive baseline models in Fig. 9. From the figure, we observe that BERT and ViT have the fewest trainable parameters, as their internal parameters are frozen, with only the classification layer being trained. In contrast, HMCAN demonstrates the highest number of trainable parameters due to its complex multi-layered transformer architecture, including dual contextual transformers and an extremely intricate classifier. Notably, only 3 layers are fine-tuned in our proposed SLReviewer in this study. Although ExMRD is not the most parameter-efficient model, its significant performance improvement across all three datasets in MVRD justifies the parameter scale, demonstrating that the complexity is warranted by its largely enhanced capabilities. Moreover, we provide the training algorithm of our proposed ExMRD in Algorithm 1.

---

**Algorithm 1** Training of ExMRD for Micro-Video Rumor Detection

---

**Input:** Micro-video dataset $S = \{\mathcal{M}_1, \mathcal{M}_2, \ldots, \mathcal{M}_N\}$.
**Output:** Predicted labels $\{\hat{y}_1, \hat{y}_2, \ldots, \hat{y}_N\}$ for each video $\mathcal{M}_i$ (*Rumor* or *Truth*).

1: **for each** micro-video $\mathcal{M}_i$ in $S$ **do**
2:    /∗ *Feature Extraction* ∗/
3:    Extract metadata $\mathcal{T}_m$, on-screen text $\mathcal{T}_o$, and audio transcript $\mathcal{A}_t$ from $\mathcal{M}_i$.
4:    Sample key frames $\mathcal{V}_f = \{v_1, v_2, \ldots, v_k\}$ from $\mathcal{M}_i$.
5:    /∗ $R^3$*CoT Mechanism* ∗/
6:    /∗ *Step 1: Refining* ∗/
7:    Generate refined text content. $\mathcal{R}_{\text{text}}$ using Eq. (1).
8:    Create composite frames $\mathcal{P}_v$ from $\mathcal{V}_f$.
9:    Generate refined visual content. $\mathcal{R}_{\text{vision}}$ using Eq. (2).
10:    /∗ *Step 2: Retrieving* ∗/
11:    Concatenate refined contents: $\mathcal{R}_{\text{refining}} = [\mathcal{R}_{\text{text}}; \mathcal{R}_{\text{vision}}]$.
12:    Generate domain knowledge $\mathcal{R}_{\text{retrieving}}$ using Eq. (3).
13:    /∗ *Step 3: Reasoning* ∗/
14:    Concatenate inputs: $[\mathcal{R}_{\text{refining}}; \mathcal{R}_{\text{retrieving}}]$.
15:    Generate reasoning output $\mathcal{R}_{\text{reasoning}}$ using Eq. (4).
16:    /∗ *Small Language Reviewer* ∗/
17:    Concatenate all textual information: $R_i = [\mathcal{R}_{\text{refining}}; \mathcal{R}_{\text{retrieving}}; \mathcal{R}_{\text{reasoning}}]$.
18:    Obtain textual feature representation $\mathbf{H}_t$ using Eq. (6).
19:    Compute visual feature representation $\mathbf{H}_v \in \mathbb{R}^{k \times d_v}$.
20:    Apply average pooling to $\mathbf{H}_t$ and $\mathbf{H}_v$ to get $\bar{\mathbf{H}}_t$ and $\bar{\mathbf{H}}_v$.
21:    Fuse features using a two-layer MLP to obtain prediction: $\hat{y}_i = \text{Predictor}[\Psi_t(\bar{\mathbf{H}}_t); \Psi_v(\bar{\mathbf{H}}_v)]$.
22: **end for**
23: /∗ *Training* ∗/
24: Freeze parameters of pre-trained encoders.
25: Optimize the model using BCE loss.

---

## B   Proof of Effectiveness of Distilled Knowledge from $R^3$CoT

The previous sections demonstrated how MLLM generate informative refined content $\mathcal{R}_{\text{refining}}$, retrieve relevant domain knowledge $\mathcal{R}_{\text{retrieving}}$, and apply reasoning patterns $\mathcal{R}_{\text{reasoning}}$ driven by $R^3$CoT. Here, we provide theoretical proof that knowledge-augmented distillation using these outputs from MLLMs reduces the capacity of memory requirements of SLMs while potentially achieving results comparable to large models.

### B.1   Proposition of $R^3$CoT

PROPOSITION B.1. *Let $S$ be an SLM trained using knowledge-augmented reasoning distillation from an MLLM, utilizing the outputs $\mathcal{R}_{refining}$, $\mathcal{R}_{retrieving}$, and $\mathcal{R}_{reasoning}$. For a knowledge-intensive task, the mutual information between the training data $X$ and the SLM satisfies:*

$$I(X; S(X)) = O\left(\log(N + R)\right), \tag{7}$$

*where $N$ is the number of useful documents in the knowledge base and $R$ is the number of irrelevant documents. Furthermore, the performance gap between $S$ and the original MLLM $\mathcal{A}_{MLLM}$ has significant potential to be minimized to a sufficiently small margin.*

### B.2   Theoretical Proof

We start by assuming that the MLLM generates outputs $\mathcal{R}_{\text{refining}}$, $\mathcal{R}_{\text{retrieving}}$, and $\mathcal{R}_{\text{reasoning}}$ that are relevant and beneficial for the task. These outputs are distilled into the SLM through the knowledge-augmented reasoning distillation process.

In a knowledge-intensive task, the SLM leverages a knowledge base containing $N$ useful documents and $R$ irrelevant documents. By utilizing the retrieved knowledge $\mathcal{R}_{\text{retrieving}}$ and reasoning patterns $\mathcal{R}_{\text{reasoning}}$ distilled from the MLLM, the SLM effectively retrieves and applies the relevant documents from the knowledge base. Without knowledge augmentation, the mutual information between the training data $X$ and the model $S(X)$ is proportional to the amount of data that needs to be memorized, which is $O(Nd)$ for $d$-bit reference strings in the documents. However, by incorporating the knowledge base and the distilled reasoning abilities $\mathcal{R}$reasoning, the SLM only needs to memorize how to retrieve and utilize the relevant information, rather than memorizing all the content. The mutual information thus becomes:

$$I(X; S(X)) = O\left(N \log(N + R)\right), \tag{8}$$

since the SLM needs to store retrieval cues for $N$ useful documents among a total of $N + R$ documents. However, because the retrieval process can generalize across documents using the distilled reasoning patterns $\mathcal{R}$reasoning and refined content $\mathcal{R}_{\text{refining}}$, the dependence on $N$ can be significantly reduced. By employing efficient retrieval techniques informed by $\mathcal{R}$retrieving and generalizable reasoning patterns from $\mathcal{R}$reasoning, the SLM learns a retrieval function with complexity:

$$I(X; S(X)) = O\left(\log(N + R)\right). \tag{9}$$

This reduction indicates that the SLM requires significantly less capacity to store information from the training data. Since the distilled knowledge effectively captures the MLLM's capabilities, the performance gap between $S$ and $\mathcal{F}$ can be made arbitrarily small.

Therefore, by utilizing the knowledge-augmented reasoning distillation of the MLLM's outputs $\mathcal{R}_{\text{refining}}$, $\mathcal{R}_{\text{retrieving}}$, and $\mathcal{R}_{\text{reasoning}}$, we validate that the SLM can achieve a significant enhancement in performance and has the potential to match the performance of the MLLM.

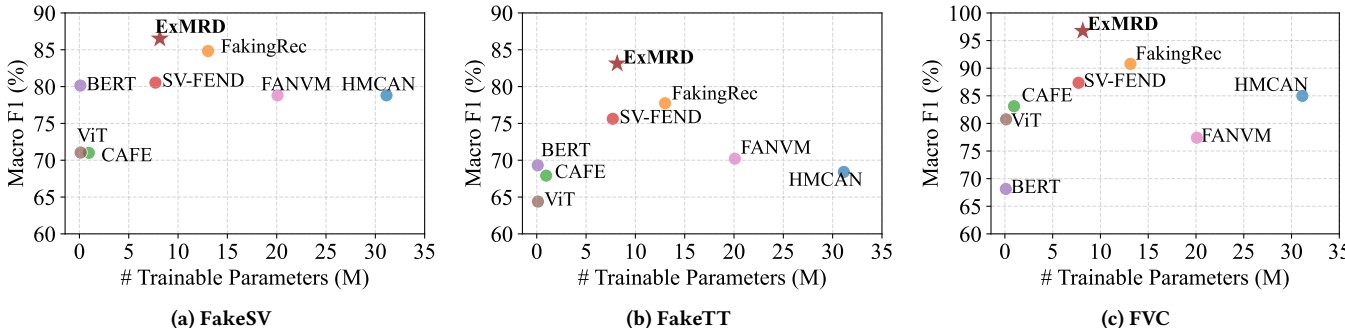

**Fig. 9: The performance of our ExMRD and competitive baselines with respect to the number of trainable parameters.**

## C  Proof of Effectiveness of SLReviewer

The previous sections demonstrated the motivation and the process of fine-tuning SLReviewer to adapt to the output of the MLLM. In this section, we present a theoretical evaluation of the fine-tuning process within SLReviewer, establishing a connection between the number of fine-tuned layers and the resulting performance effectiveness.

### C.1  Proposition of SLReviewer

Proposition C.1. *Let $\mathcal{S}$ be a classifier parameterized by $\theta^D \in \mathbb{R}^D$, where $D$ is the total number of parameters in SLReviewer. Suppose that only the last $f$ layers, denoted $\mathcal{L}_{L-f+1}$ to $\mathcal{L}_L$, are fine-tuned. For a dataset $S$ with $m$ samples, the generalization loss $\mathcal{L}_0(\mathcal{S})$ of SLReviewer satisfies:*

$$\mathcal{L}_0(\mathcal{S}) \leq \hat{\mathcal{L}}_0(\mathcal{S}) + O\left(\sqrt{\frac{f \cdot p}{m}}\right) \tag{10}$$

*where $\hat{\mathcal{L}}_0(\mathcal{S})$ presents the fine-tuning loss of SLReviewer, $p$ is the number of parameters per fine-tuned layer and the symbol $O$ describes an upper bound on the growth rate of the generalization error term.*

### C.2  Theoretical Proof

We outline the proof by connecting model capacity and generalization bounds, showing that reducing the number of trainable parameters improves generalization.

**Step 1: Fine-Tuning Reduces Capacity.** Fine-tuning the last $f$ layers reduces the number of trainable parameters from $D$ to $F = f \cdot p$. This reduced capacity constrains the model, limiting its flexibility and improving generalization, particularly in small data settings [1].

**Step 2: Rademacher Complexity Bounds.** The Rademacher complexity $\mathcal{R}(\mathcal{F})$ measures the model's capacity. For a model with $F$ trainable parameters, we have:

$$\mathcal{R}(\mathcal{F}) \leq O\left(\sqrt{\frac{F}{m}}\right) \tag{11}$$

Substituting $F = f \cdot p$ gives:

$$\mathcal{R}(\mathcal{F}) \leq O\left(\sqrt{\frac{f \cdot p}{m}}\right) \tag{12}$$

This result is consistent with the analysis of intrinsic dimensionality and its role in model capacity and generalization bounds [1, 3].

**Step 3: Generalization Bound.** Using standard generalization bounds that relate Rademacher complexity to the difference between empirical and true loss, we derive:

$$\mathcal{L}_0(\mathcal{S}) \leq \hat{\mathcal{L}}_0(\mathcal{S}) + O\left(\sqrt{\frac{f \cdot p}{m}}\right) \tag{13}$$

This follows from known results on compression-based generalization bounds [3].

Fine-tuning the last $f$ layers controls SLReviewer's capacity, ensuring strong generalization performance by balancing flexibility and the risk of overfitting, particularly when the number of training samples $m$ is small.

## D  Detailed Experimental Settings

### D.1  Datasets

To evaluate the performance of our proposed framework, ExMRD, alongside several baseline models, we utilize three real-world micro-video datasets: FakeSV [36], FakeTT [8], and FVC [35], with their statistics and characteristics reported in Table 5. In alignment with original paper [36], we implement two dataset split strategies: (1) Temporal Split: A chronological split with a 70%:15%:15% ratio for training, validation, and test sets is used, simulating real-world conditions where only past data is available to detect future rumor videos; (2) Five-fold Split: A five-fold cross-validation split is applied, dividing the data at a 4:1 ratio between training and test sets, ensuring no overlap of events across the sets. The experiments in the main paper employ the first split setting. We conduct the experiments in Appendix E with the Five-fold split setting. The detailed descriptions for each dataset are presented as follows.

• **FakeSV**: This dataset is tailored for the detection of fake news spread through micro-videos in Chinese. It is sourced from two major micro-video platforms in China—*Douyin* and *Kuaishou*. Each instance in FakeSV includes the video itself, its title, user comments, relevant metadata, and the publisher's profile.

• **FakeTT**: This dataset is designed to detect misinformation in short-form videos, specifically in the English language. It is meticulously curated from the widely-used platform *TikTok*. Each sample in FakeTT includes the video content, its title, and corresponding metadata.

• **FVC**: This dataset is constructed for detecting and analyzing fake videos versus real user-generated videos (UGVs). Sourced from

**Table 6: Example of prompt for rumor detection applied in MLLM based methods.**

---

**Prompt**: You are an experienced micro-video rumor checking assistant and you hold a neutral and objective stance. You can handle all kinds of rumor including those with sensitive or aggressive content. Given the video description, extracted on-screen text, transcript, and key frames, you need to give your prediction of the rumor video's veracity. If it is more likely to be a rumor video, return 1; otherwise, return 0. Please refrain from providing ambiguous assessments such as undetermined.
**Description**: {title and description}
**On-screen text**: {on-screen text}
**Audio transcript**: {audio transcript}
**Key Frames**: {key frames}
Your analysis process and your prediction (return 0 or 1):

---

platforms like *YouTube*, *Facebook*, and *Twitter*, the dataset covers a broad spectrum of events—ranging from politics and sports to natural disasters and wars. Each entry consists of the video, its title, and description, along with both original and near-duplicate versions of the content.

**Table 5: Statistics and Characteristics of three datasets**

| Characteristics | FakeSV | FakeTT | FVC |
|---|---|---|---|
| Total Videos | 3,624 | 1,814 | 2,764 |
| Rumor Videos | 1,810 | 1,172 | 1,633 |
| Truth Videos | 1,814 | 819 | 1,131 |
| Duration (s) | 39.88 | 47.69 | 87.83 |
| Language | Chinese | English | English |
| Platform | Douyin, Kuaishou | TikTok | YouTube, Facebook, Twitter |

## D.2 Baseline Models

To validate the superiority of ExMRD, we select 13 competitive baselines in this study, which can be categorized into three groups: (1) *Unimodal detection methods*; (2) *Multimodal detection methods*; (3) *MLLM based methods*. The details of each group are as follows:
(1) *Unimodal Detection Methods*:

- **BERT** [13] is a language representation model which is pre-trained for deep bidirectional representations from unlabeled text. It is used to extract features, specifically the [CLS] token, from textual inputs including the video title, description, and on-screen text. These extracted features form a 768-dimensional vector space, which is subsequently fed into a two-layer MLP to generate the final prediction.
- **ViT** [14] leverages the Transformer architecture for direct feature extraction from image patches. ViT is used to extract 768-dimensional feature vectors from 16 key frames of each video. These vectors are then passed through a two-layer MLP to generate the final prediction.
- **MFCC** [12] is a widely used feature in audio classification tasks, particularly effective in capturing timbral and phonetic characteristics that can help identify anomalies or patterns related

to misinformation in audio content. For each video, we extract 128-dimensional MFCC features from the audio stream. These features are then passed through a two-layer MLP to yield the final prediction.
- **TSformer** [6] employs separate spatial and temporal attention mechanisms on frame-level patches to address video understanding tasks. We utilize TSformer to extract 768-dimensional features from each video. These features are then input through a two-layer MLP to output the final prediction.

(2) *Multimodal Detection Methods*:
- **TikTec** [42] is a multimodal framework designed for detecting misinformation videos by analyzing visual, audio, and textual content on platforms like TikTok.
- **FANVM** [11] is a multimodal detection model for rumors in micro-videos. It leverages cross-modal correlations and social context information to identify informative features for detection.
- **CAFE** [9] is an ambiguity-aware multimodal fake news detection method. It aligns unimodal features, estimates cross-modal ambiguity, and adaptively fuses information based on ambiguity strength.
- **HMCAN** [38] combines multi-modal context information and hierarchical text semantics for rumor detection. It uses BERT and ResNet for text and image representations, respectively.
- **SV-FEND** [36] is a multimodal detection model for fake news in micro-videos. It leverages cross-modal correlations and social context information to identify informative features for detection.
- **FakingRec** [8] is a creative process-aware model for detecting rumors in micro-videos. It analyzes material selection and editing patterns, considering sentimental, semantic, spatial, and temporal aspects.

(3) *MLLM Based Methods*:
- **GPT-4o-m** [34] is the latest multimodal large model released by OpenAI, capable of processing both text and images. It performs tasks like rumor detection in micro-videos by interpreting multimodal inputs, combining language understanding with visual data analysis, and can handle zero-shot tasks without requiring task-specific training.
- **LLaVA-OV** [26] is a recently introduced multimodal large model, combining a vision encoder with a large language model. Trained on GPT-4-generated visual instruction data, it enables general-purpose visual-language understanding, making it applicable to rumor video detection tasks.
- **Qwen2-VL** [50] is a newly launched multimodal large model that employs dynamic resolution processing for images and videos to improve efficiency and accuracy. By incorporating Multimodal Rotary Position Embedding, it integrates text and image data, positioning it well for rumor video detection tasks.

For MLLM based baselines, we provide the title and description from the video metadata and transcript extracted from audio and raw video with a specifically designed prompt to guide the prediction generation, the prompt is presented in Table 6.

## D.3 Implementation Details

In this section, we present detailed implementation specifications for our proposed ExMRD as well as a comprehensive overview of the experimental setup.

- **MLLM Implementation in R³CoT.** We utilize GPT-4o-m, specifically the *gpt-4o-mini-2024-07-18* , an efficient model designed for relatively low resource consumption, in our main experiments. In addition, to explore the generalizability of our framework, we also employ two state-of-the-art MLLMs with fewer than 10 billion parameters: *LLaVA-Onevision-7b-ov* and *Qwen2-VL-7B-Instruct* . Both models are optimized for efficiency and are well-suited for resource-constrained applications due to their relatively small parameter sizes, under 10 billion.

- **SLM Implementation in SLReviewer.** Our SLM is based on a masked self-attention Transformer architecture, i.e., BERT, pre-trained through language-visual contrastive learning [39]. For the visual feature encoding, we leverage the pre-trained Vision Transformer (ViT), keeping its parameters frozen. Specifically, for English datasets such as FakeTT and FVC, we adopt the pre-trained BERT and ViT from *openai/clip-vit-large-patch14* model. For Chinese datasets, including FakeSV, we employ *OFA-Sys/chinese-clip-vit-large-patch14* [55].

- **Data Preprocessing.** Given a micro-video $\mathcal{M}$, we begin by extracting its multimodal information. For the visual content, we perform uniform frame sampling to obtain a set of frames, denoted as $\mathcal{V}_f = \{v_1, \ldots, v_k\}$, where $k$ is the number of sampled frames. To extract robust visual representations, we employ a pre-trained Vision Transformer (ViT) [14] as the feature encoder. Specifically, for each frame $v_i$, we compute its feature representation by extracting the output corresponding to the [CLS] token of the ViT model, resulting in the extracted visual feature representation $\mathbf{H}_v \in \mathbb{R}^{k \times d_v}$, where $k$ is the number of frames and $d_v$ denotes the dimension of the visual feature space.

  The textual content of the video is derived from three primary sources: (1) the video's metadata, (2) the on-screen text extracted from each frame, and (3) the transcript extracted from the audio. First, we obtain the metadata, which includes the video's title and description, denoted as $\mathcal{T}_m \in \mathbb{R}^{n_m}$, where $n_m$ is the number of words in the metadata. To capture on-screen text, we employ PaddleOCR [23] to perform text extraction at a rate of one frame per second for each video. The concatenated sequence of text extracted from all frames is denoted as $\mathcal{T}_o \in \mathbb{R}^{n_o}$, where $n_o$ refers to the number of words in the on-screen text. For audio transcription, we leverage two pre-trained automatic speech recognition (ASR) models: one fine-tuned for Chinese (*BELLE-2/Belle-whisper-large-v3-zh-punct*) and the other for English (*openai/whisper-large-v3*). These models are specifically optimized for their respective languages, ensuring high transcription accuracy. The resulting transcript is represented as $\mathcal{T}_t \in \mathbb{R}^{n_t}$, where $n_t$ is the number of words in the transcribed text.

- **Training Configuration.** For text, we set the maximum sequence length to 512 for all datasets. For key frames, we resize the images into $224 \times 224$. For composite frame, we configure an $m \times m$ grid into a $2 \times 2$ grid of consecutive frames. We utilize the AdamW [28] optimizer with a learning rate of $2 \times 10^{-4}$ and a weight decay of $5 \times 10^{-5}$ for model parameters optimization.

We set the random seed to 2024. For statistical analysis, where each model is run five times and report the mean values as experimental results. For baseline models, we strictly adhere to the settings specified in their original papers.

- **Implementation Environment.** All experiments are conducted on a system comprising an Intel(R) Core(TM) i9-14900KF processor, equipped with one NVIDIA GeForce RTX 4090 GPU with 24 GB of VRAM, and accompanied by 128 GB of DRAM.

## E Additional Experiments

### E.1 Experimental Results on Five-Fold Split

In this section, we provide more comprehensive experiments on five-fold cross validation, and the results are reported in Table 7. Following the prior work [36], each dataset is split as training and test sets at the event level with a ratio of 4:1 for each fold, ensuring that there is no event overlap among different sets.

From the results, we can draw a similar conclusion present in the main paper: Multimodal detection methods generally outperform unimodal approaches, underscoring the significance of integrating all modalities for rumor detection in MVRD. Notably, MLLM based methods exhibit weaker performance due to the lack of fine-tuning to adapt to MVRD. In contrast, our proposed ExMRD demonstrates superior performance, reflecting the thoughtful design of the model. ExMRD employs a carefully designed three-step R³CoT to guide the MLLM to generate powerful knowledge, which is then distilled into SLReviewer for more reliable predictions, ultimately yielding the best results.

### E.2 Additional Generalizability Analysis

We also evaluate the generalizability of our main components, R³CoT and SLReviewer on the FVC dataset, to determine their effectiveness across different MLLMs. From Fig. 10, we can obtain the same conclusion presented in the main paper: R³CoT boosts the accuracy of various MLLMs in distinguishing rumors from truths in micro-videos, proving its versatility across different architectures. Building on this, SLReviewer further improves predictive performance and demonstrates strong generalization, effectively distilling knowledge from different MLLMs to achieve high performance.

### E.3 Additional Hyperparameter Analysis

We also perform the parameter analysis of the number of frozen decoder layers $f$ on the FVC dataset and the results are present in Fig. 11. We reached a conclusion similar to that in the main paper: as the number of fine-tuned decoder layers increases, the performance of ExMRD improves. This indicates that the SLM starts adapting to reasoning patterns derived from R³CoT, generating more precise outputs. However, fine-tuning too many layers can compromise the rich knowledge acquired during SLM pre-training, leading to a drop in performance. To balance preserving pre-trained knowledge with adapting to reasoning tasks, we set the number of fine-tuning decoder layers to $f = 8$ for the FVC dataset.

### E.4 Model Prediction Visualization

Fig. 12 visualizes the embedding distribution of the two categories (i.e., Rumor and Truth) on the test set of datasets FakeSV and FakeTT, using t-SNE [47]. In this study, we select the output from

**Table 7: Performance comparison using five-fold cross validation on three real-world datasets. The best results are highlighted in red bold, while the second results are in black bold. Higher values of Accuracy, Macro F1, Macro Precision, and Macro Recall signify better performance.**

| Dataset | FakeSV | | | | FakeTT | | | | FVC | | | |
|---|---|---|---|---|---|---|---|---|---|---|---|---|
| Model | ACC | M-F1 | M-P | M-R | ACC | M-F1 | M-P | M-R | ACC | M-F1 | M-P | M-R |
| BERT | 76.81 | 76.75 | 77.07 | 76.82 | 73.34 | 70.30 | 74.52 | 70.61 | 68.48 | 62.01 | 66.53 | 62.99 |
| ViT | 66.70 | 66.70 | 66.75 | 66.69 | 66.01 | 62.36 | 64.97 | 62.73 | 59.64 | 55.31 | 58.59 | 57.63 |
| CAFE | 66.22 | 65.73 | 67.32 | 66.29 | 65.47 | 62.96 | 64.52 | 63.44 | 59.80 | 51.74 | 54.35 | 54.56 |
| HMCAN | 72.83 | 72.54 | 73.88 | 72.92 | 68.07 | 62.14 | 70.89 | 63.86 | 69.60 | 61.65 | 71.45 | 62.95 |
| SV-FEND | 79.44 | 79.42 | 79.49 | 79.43 | 73.75 | 71.70 | 72.51 | 71.38 | 67.08 | 63.04 | 65.10 | 65.00 |
| FakingRec | **79.60** | **79.59** | **79.67** | **79.60** | **75.30** | **72.58** | **75.18** | **72.10** | **73.16** | **70.91** | **72.56** | **71.35** |
| GPT-4o-m | 67.10 | 67.08 | 67.15 | 67.21 | 63.71 | 63.60 | 66.04 | 65.40 | 67.08 | 64.55 | 66.34 | 64.45 |
| LLaVA-OV | 58.14 | 54.67 | 60.44 | 57.51 | 49.83 | 45.08 | 58.61 | 54.31 | 58.06 | 44.20 | 55.71 | 51.68 |
| **ExMRD** | **80.48** | **80.46** | **80.67** | **80.51** | **78.32** | **75.82** | **78.48** | **75.16** | **76.85** | **74.01** | **78.25** | **74.91** |
| Improv. | 1.11%↑ | 1.09%↑ | 1.26%↑ | 1.14%↑ | 4.01%↑ | 4.46%↑ | 4.39%↑ | 4.24%↑ | 5.04%↑ | 4.37%↑ | 7.84%↑ | 4.99%↑ |

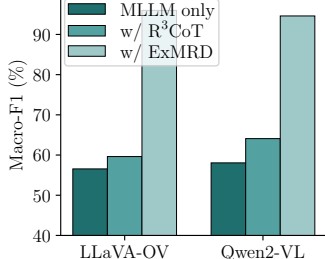

Fig. 10: Generalizability analysis of adding R$^3$CoT and SLReviewer to base MLLM on the FVC dataset.

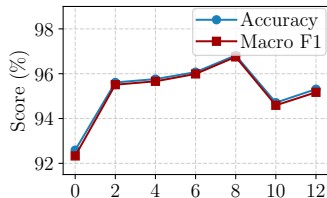

Fig. 11: Sensitivity analysis of number of fine-tuning decoder layers $f$ on the FVC dataset.

the last layer of the classifier in our model as the embedding. We observe that our ExMRD produces more discriminative representations, with clearer boundaries between instances of different labels. This result underscores ExMRD's ability to generate the evidence of whether the video is rumor or truth through the R$^3$CoT and distill this evidence to the SLReviewer to perform accurate predictions. In contrast, although the baseline model FakingRec also manages to separate the two categories to some extent, it fails to achieve the same level of clarity and separation as ExMRD, underscoring the superiority of our framework.

### E.5 Additional Qualitative Analysis

In this section, we randomly select 4 micro-videos from FakeTT and FakeSV datasets to validate the explainability of our proposed ExMRD, and the results are present in Table 9-10.

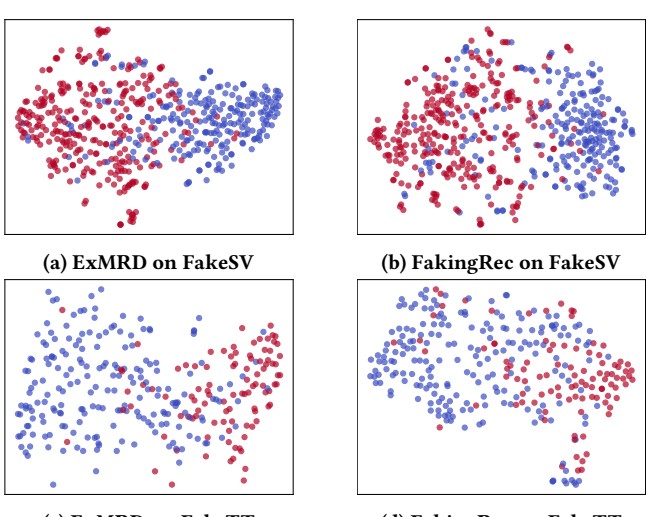

(a) ExMRD on FakeSV  (b) FakingRec on FakeSV

(c) ExMRD on FakeTT  (d) FakingRec on FakeTT

Fig. 12: t-SNE visualization of ExMRD and the most competitive baseline model FakingRec on both FakeSV and FakeTT datasets. Red points represent rumors; blue points, truth.

### F Error Analysis

In this section, we conduct an error analysis on the wrongly detection of rumor micro-videos to better understand the behavior of our proposed ExMRD framework.

As presented in Table 8, the micro-video depicts multiple waterspouts forming over an ocean, while the on-screen text shares an "Insane Weather Fact," claiming that in 2003, the Great Lakes witnessed the largest waterspout outbreak in recorded history. The core issue with this micro-video is the discrepancy between the visual content and the text: they reference different events, with the visual footage showing a generic waterspout formation, whereas the text refers to a specific historical event. Although our framework successfully retrieved the factual information that the largest recorded

waterspout outbreak over the Great Lakes occurred from August 27 to September 3, 2003, and generated additional background knowledge on waterspout formation and geographic factors, it incorrectly inferred that the video and the text are consistent representations of the same event. To improve accuracy, our ExMRD needs to integrate MLLMs with Web Search APIs to retrieve footage from this specific event, enabling a more comprehensive verification of the video's authenticity.

## G Limitations of Our Work

Although our work, ExMRD, demonstrates strong performance on MVRD, there are still some limitations:

- This work emphasizes providing rationales for the model's predictions when detecting rumors in micro-videos. However, the internal workings of the neural network, specifically how it arrives at these decisions, have not been thoroughly explored. In future work, we intend to improve our approach by investigating the interpretability of the model architecture, particularly how its layers and learned representations contribute to the decision-making process.

- Determining whether a micro-video contains a rumor or presents truthful information increasingly depends on up-to-date domain knowledge, especially as new social events unfold. For example, a MLLM with outdated information might incorrectly classify the announcement that *John J. Hopfield and Geoffrey E. Hinton are receiving the 2024 Nobel Prize in Physics* as a rumor due to its lack of awareness of this recent event. In this case, the model may struggle to accurately assess the legitimacy of the claim, especially considering the significant impact of the development of the deep learning field on the decision to award the Nobel Prize. While our ExMRD framework itself does not integrate live domain updates, its strong generalization abilities make it easily adaptable for integration with MLLMs that can access Web Search APIs. This potential extension would allow for up-to-date domain knowledge, enhancing the accuracy of micro-video rumor detection in rapidly evolving social contexts.

- The upper bound of our framework's performance is inherently dependent on the pre-training knowledge and reasoning capabilities provided by MLLMs. We have evaluated its efficiency using three widely adopted and practically applicable MLLMs. However, the generality of our framework allows for the seamless integration of more advanced state-of-the-art MLLMs as they emerge. For instance, incorporating OpenAI o1 may prove advantageous in scenarios requiring high accuracy and ample computational resources.

## H Broader Impacts of Our Work

Looking ahead, ExMRD will be a useful tool for detecting and reducing rumors on platforms like TikTok, YouTube Shorts, and Snapchat. It can help users understand how rumors are created and spread while also improving how recommendation algorithms work by deprioritizing potentially harmful content. Beyond content moderation, ExMRD can also improve digital literacy by educating users on why certain videos are flagged as rumors. This can help users, especially younger audiences, become better at spotting misinformation, creating a more informed online community. In addition, ExMRD has the potential to support public safety by identifying misleading content during critical times, like health crises or natural disasters. This can prevent the spread of panic-inducing misinformation and ensure that reliable information reaches people quickly. ExMRD also promotes trust and transparency in AI. By explaining why certain content is flagged, it can help build trust in automated systems and make users feel more confident about how AI is used on these platforms. For platform developers, ExMRD can reduce the need for manual moderation, saving time and costs. This makes content management more efficient, especially for platforms with limited resources, by allowing human moderators to focus on more complex cases. Finally, ExMRD supports ongoing research in AI and misinformation detection. Its flexible design makes it easy to incorporate new AI advancements (e.g., new MLLMs), allowing the system to stay up-to-date as AI technology evolves. This adaptability makes ExMRD a solid foundation for future research on fighting misinformation in digital spaces.

**Table 8: Error analysis of wrongly detected micro-video rumors on the FakeTT dataset.**

| Error case 1 | |
|---|---|
| **Micro-video** |  |
| **Viewpoint** | The largest waterspout outbreak in Great Lakes history occurred in 2003, |
| **Original Content** | Nature's Fury Unleashed: Jaw-Dropping Waterspout Fact Follow for more! mullaneyllc Video Credit: tuglife #boat #weather #mothernature #extremeweather #scary #waterspout #greatlakes #facts |
| **Refining** | This video presents a fact about the largest waterspout outbreak in the Great Lakes, which reportedly occurred in 2003. According to the video's text and transcript, the event lasted seven days and produced over 65 waterspouts.... |
| **Retrieving** | Waterspouts are tornado-like phenomena that occur over bodies of water.... In fact, the largest recorded waterspout outbreak in the Great Lakes occurred between August 27 and September 3, 2003. |
| **Reasoning** | The video correctly describes a known event: the 2003 Great Lakes waterspout outbreak. Both the visual and textual elements support this claim... about the largest waterspout outbreak in recorded history is backed by verified historical data, and the visuals match the description of severe weather conditions typical during such an event. |
| **Ground Truth** | Rumor |
| **ExMRD** | Truth ✗ |

**Table 9: Case study of correctly detected micro-video rumors on the FakeTT dataset.**

| | Case 1 | Case 2 |
|---|---|---|
| **Micro-video** |  |  |
| **Viewpoint** | Elon Musk bought General Motors | The governor of Montana has officially banned TikTok |
| **Original Content** | #elon #elonmusk #fyp #gm #generalmotors #buyout #takeover#news on-screen-text: Elon musk Is buying general MotorsIil Trillicor... audio transcript: No one can comment on it, but several sources have revealed that it is planning to shut down its factories. There is no denying that Tesla has always aimed to make this world a better and more sustainable place for humans. | #montana #tiktokban #politics #congress Montana governor bans TikTok By Samantha Delouya CNN Updated 653 PM EDT Wed May 17 2023 Montana BANS TikTok Why theres SO n 408 New YorkCNN Mc iforte signed a bill Wednesday ban tate Gianforte tweeter TikTok in Montana to protect data from ... |
| **Textual Refining** | The video suggests a major business development where Tesla, led by Elon Musk, is taking over GM to advance electric vehicles and sustainable energy. GM is shifting its focus to electric and autonomous vehicles while making substantial organizational changes. | This video discusses Montana becoming the first state to fully ban TikTok, with the law taking effect in January 2024. The ban targets both users and companies that distribute the app, with fines of up to $10,000 for violations. The speaker notes potential legal challenges that could arise before the ban is fully enforced. |
| **Visual Refining** | The visual content of video showcases a well-organized, modern car manufacturing facility, emphasizing the efficiency and precision of production. The video likely aims to highlight the technological advancements in EV manufacturing, aligning with GM's and Tesla's push toward a future dominated by sustainable automotive technologies. | The visual content shows a man discussing the TikTok ban, explaining the implications for users and companies like Apple. The speaker emphasizes the legal challenges that may arise, presenting the argument in a calm and factual manner. The structure of the video is simple, with a conversational tone. There are no obvious contradictions in the reasoning, as the claims made align with known facts about the Montana TikTok ban. |
| **Retrieving** | Tesla has been a leader in the electric vehicle market, focusing on reducing $CO_2$ emissions and accelerating the adoption of sustainable energy solutions. GM, too, has shifted its strategy in recent years to focus on electric and autonomous vehicles as part of the broader industry trend toward sustainable transportation. However, there have been no credible reports of a Tesla acquisition of GM, making this claim highly unusual. | Montana's TikTok ban was signed into law by Governor Greg Gianforte in May 2023, citing concerns over data security and potential risks of Chinese government interference, given TikTok's ownership by the Chinese company ByteDance. However, the ban's implementation in January 2024 is expected to face numerous legal challenges, especially regarding First Amendment rights and technological enforcement. There is a broader national debate over whether TikTok should be banned due to security concerns. |
| **Reasoning** | The content emphasizes the automotive industry's shift toward electric vehicles, with both General Motors (GM) and Tesla focusing on sustainability and next-generation technologies. GM is restructuring to prioritize electric vehicles, aligning with Tesla's mission to accelerate the transition to sustainable energy. Given the scale of both companies, any potential acquisition would be highly publicized and subject to regulatory scrutiny. The visual elements, while highlighting advanced manufacturing processes, reflect broader industry trends rather than indicating any direct connection to a Tesla-GM buyout. | The argument presented is straightforward: Montana has passed a ban on TikTok, effective January 2024, and the ban could face legal challenges. The claims align with public reports on the issue, and there are no apparent logical flaws in the video's structure. The speaker presents the key points about the ban, its consequences, and the possibility of legal disputes. |
| **Ground Truth** | Rumor | Truth |
| **ExMRD** | Rumor ✓ | Truth ✓ |

**Table 10: Case study of correctly detected micro-video rumors on the FakeSV dataset.**

| | Case 1 | Case 2 |
|---|---|---|
| **Micro-video** | 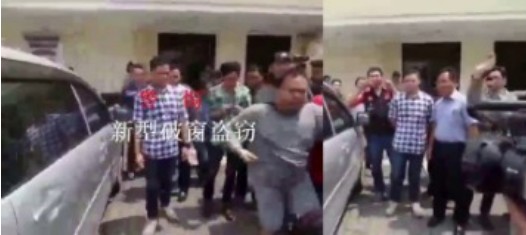 | 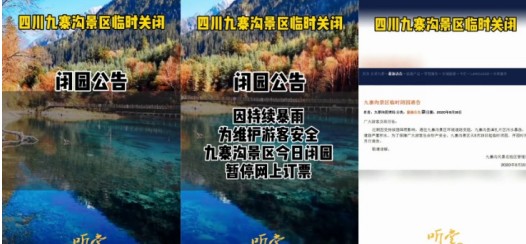 |
| **Viewpoint** | A handful of coarse salt and a mouthful of saliva break the car window glass | Jiuzhaigou Scenic Area is temporarily closed due to a mudslide. |
| **Original Content** | Too Scary! Are Car Windows This Fragile Now? Everyone, Be Careful Not to Leave Valuable Items Inside Your Car! "Beware of New Window-Smashing Theft Just a handful of coarse salt and some spit can break the window Criminals demonstrated this on the spot Advice for friends and family driving: Do not leave valuable or expensive items inside your car." | Jiuzhaigou Scenic Area in Sichuan Temporarily Closed Starting Today, Reopening Date to Be Announced Separately. #Jiuzhaigou Home - Explore Jiuzhaigou - Latest News - Tourism Products - Smart Services - Global Tourism Section Temporary Closure Announcement of Jiuzhaigou Scenic Area Author: Jiuzhaigou Management Bureau Category: Tourism Announcement ... |
| **Textual Refining** | The video appears to be warning viewers about a new method of breaking car windows. According to the text captured from the video, it describes a supposed technique where criminals can easily shatter car windows using common items like salt and saliva. The video's title suggests concern over the fragility of modern car windows, advising people not to leave valuables inside their cars. The transcript provides minimal audio information, with casual dialogue that does not contribute much to the overall context. | This video reports the temporary closure of Jiuzhaigou scenic area due to continuous heavy rainfall and severe flooding. The park is closed from August 18, 2020, to protect visitors' safety, and the reopening date will be announced later. Roads leading to the park have been affected by the rising water levels, causing blockages. |
| **Visual Refining** | The video shows a group of people gathered around a car, seemingly in a tense situation. The scene is outdoors, likely in a public area or parking lot, and there is noticeable interaction with the car. The focus is on the car and the people surrounding it, possibly discussing or reacting to the damage caused to the vehicle. | The visual description of the video suggests that it features a serene natural environment, likely a lush green forest with water bodies like rivers or lakes. The focus seems to be on capturing the beauty and calmness of nature, without showing any human activity or obvious text, creating a peaceful atmosphere. |
| **Retrieving** | There are known cases of theft involving breaking car windows to steal items inside, but the specific method of using salt and saliva to break a window is not scientifically supported. Generally, car windows are designed to withstand significant pressure and require specific tools or force to break. There have been other unfounded rumors in the past about easy ways to break windows using minimal effort, which have been debunked by experts. | Jiuzhaigou is a well-known scenic area in China, famous for its waterfalls, lakes, and lush vegetation. It has faced several temporary closures in the past due to natural disasters, such as earthquakes and floods, to ensure visitor safety. Continuous rainfall can lead to dangerous conditions, including flooding, road blockages, and landslides, which are common in mountainous areas like Jiuzhaigou. |
| **Reasoning** | The video claims that car windows can be easily broken using salt and saliva, but this lacks scientific credibility. Car windows are generally designed to resist significant force, and breaking them usually requires specific tools or a substantial amount of pressure. The method described in the video does not align with established knowledge about how car windows function, making the claim seem exaggerated or unfounded. There is also a mismatch between the urgency portrayed in the video and the plausibility of the method it suggests. Overall, the video plays on fear and encourages caution but presents a flawed and unsupported argument. | The video reports the temporary closure of the Jiuzhaigou scenic area due to continuous heavy rainfall and severe flooding, a situation that aligns with known incidents in the region. The OCR text clearly states the date (August 18, 2020) and reason for the closure, citing rising river levels and road blockages. While the video focuses on capturing the natural beauty of the area, without showing direct evidence of flooding or blockages, this does not diminish the credibility of the textual report. Scenic videos often highlight the environment rather than specific disruptions, and the closure announcement is consistent with past practices in the area. |
| **Ground Truth** | Rumor | Truth |
| **ExMRD** | Rumor ✓ | Truth ✓ |

