# OpenReview forum: "Following Clues, Approaching the Truth: Explainable Micro-Video Rumor Detection via Chain-of-Thought Reasoning"
_ACM.org/TheWebConf/2025/Conference — WWW 2025 Poster_

### Official Review · Reviewer_wdZ1 · 2024-11-25

**Novelty:** 4
**Technical Quality:** 3

**Review:**

This paper addresses the challenge of detecting rumors in micro-video content on platforms like TikTok and YouTube Shorts, where misinformation can spread rapidly. The authors introduce ExMRD, an explainable framework for Micro-Video Rumor Detection (MVRD) that combines multimodal data (text, audio, visual) with a Chain-of-Thought (CoT) reasoning mechanism named R³CoT. This mechanism involves three steps: Refining the original content, Retrieving relevant domain knowledge, and Reasoning to validate the information. The framework aims to not only improve detection accuracy but also provide clear explanations for classification decisions, enhancing user trust. The authors propose a Small Language Reviewer (SLReviewer) to streamline predictions while reducing computational costs. Experiments on real-world datasets demonstrate that ExMRD significantly outperforms existing methods in terms of accuracy and provides well-reasoned rationales for its predictions.

**Questions:**

1. Could the authors provide more detailed explanations of the R³CoT mechanism's individual steps? Specifically, how are the refining and retrieving processes operationalized in practice?
2. What criteria were used for selecting the datasets utilized in the experiments? Are there any biases or limitations in the datasets that could affect the generalizability of the results?
3. How do the authors quantitatively assess the explainability of the MVRD predictions? Are there specific metrics or user studies conducted to evaluate the effectiveness of the explanations provided by ExMRD?
4. While the paper mentions significant improvements over competitive baselines, could the authors elaborate on the specific weaknesses of these baseline methods that ExMRD addresses?

**Reviewer Confidence:**

3: The reviewer is confident but not certain that the evaluation is correct

**Scope:**

3: The work is somewhat relevant to the Web and to the track, and is of narrow interest to a sub-community

---

### Official Review · Reviewer_YXhj · 2024-11-28

**Novelty:** 4
**Technical Quality:** 5

**Review:**

This manuscript addresses the task of Micro-Video Rumor Detection (MVRD) by introducing a novel framework called ExMRD, which is designed to generate detailed and coherent explanations for rumor detection in micro-videos. The framework's core is the R^3CoT mechanism, including Refining, Retrieving, and Reasoning. To finetune MLLMs, the authors propose a Small Language Reviewer (SLReviewer) to distill the outputs of the R^3CoT-guided MLLMs for efficient and reliable predictions. The manuscript presents extensive experiments demonstrating the effectiveness of the proposed method.

Pros:
1. The R3CoT mechanism is a novel approach that addresses the challenges of low-quality video content and the need for domain knowledge and reasoning in rumor detection.
2. The manuscript includes a wide range of experiments, including ablation studies, hyperparameter sensitivity analysis, and generalizability analysis, which thoroughly evaluate the proposed method.
3. The use of the SLReviewer to distill the outputs of the MLLMs ensures efficient and reliable predictions, making the framework practical for real-world applications.

Cons
1. The manuscript does not provide sufficient evidence or experiments to demonstrate the effectiveness of the method on rumors that rely more on visual cues or behavioral patterns rather than textual content.
2. The potential issue of outdated information from MLLMs is mentioned but not fully addressed. There is no mechanism proposed to update or cross-verify the retrieved knowledge with real-time data sources.
3. The manuscript does not fully explore the interaction between textual and visual refining. There is no discussion on how the model handles cases where these refinements conflict.
4. The necessity of the SLReviewer is not fully discussed. It is unclear whether the final reasoning step alone could be sufficient for prediction, and there is no comparison with using a larger model for the final prediction.

**Questions:**

1. The manuscript claims R3CoT works well for knowledge-based rumors. How does it handle rumors based on visual cues or behavioral patterns, like human actions or speech styles? Can you provide more evidence or experiments showing its effectiveness in these cases?
2. The authors mentioned outdated information from MLLMs, like the case in Table 9 where the knowledge is outdated. Is there a mechanism in R^3CoT to handle this, or is this a current industry challenge?
3. The case studies show significant overlap between textual and video refinements. What roles do these refinements play in R^3CoT, and how much do they contribute? The ablation experiments do not fully address this.
4. The authors didn't discuss the interaction between textual and visual refining, merely concatenating the results. Have you considered scenarios where these refinements conflict? Would introducing a conflict resolution stage to reconcile conflicting information from textual and visual refinements improve both theoretical and experimental outcomes?
5. . What roles the three stages of R3CoT play, and could the final reasoning step alone be sufficient? If the SLReviewer were replaced with a larger model, how would performance differ?

**Reviewer Confidence:**

3: The reviewer is confident but not certain that the evaluation is correct

**Scope:**

4: The work is relevant to the Web and to the track, and is of broad interest to the community

---

### Official Review · Reviewer_yepA · 2024-12-01

**Novelty:** 6
**Technical Quality:** 6

**Review:**

This work details important work developing stronger methods to detect rumors in micro-video content, an issue that is increasingly important as content creation leans further into micro-videos. This work is well-motivated and its significance is clear as there are tangible offline impacts of misinformation. I am somewhat limited in my familiarity with work of this kind, however I have provided a few suggestions for details that can be added and other points of confusion.

**Pros:**
* Significant work with tangible offline impact
* To my knowledge, the analysis is high quality and demonstrates on many levels the improvement provided by the authors' contributed mechanism.
* The problem of micro-video rumor detection is not novel, but the methodology applied to this context is.

**Cons**:
* The paper does not provide discussion about how this work can be used in practice. Do the authors envision building a system that deploys their model? How would it be used by users or platforms? Because the authors mention deployment, I would like to see more detail on the practical usage of this contribution.
* There is no discussion of the ethical implications associated with this work. What are the costs associated with mis-predictions in this space? I think this work is important enough to overcome any sort of ethical concerns, but I would like to see the authors acknowledge and discuss relevant concerns.
* There is room for more discussion in the related work section regarding other work looking at micro videos generally. I suspect this isn't the first work turning micro-videos (or even regular videos) into text, but I want to know how other researchers have done it and why this work does so differently. Section 4 has a description of the experiments which references many other baselines that seem to tackle similar problems, maybe adding more detail on these in the related work would solve this confusion for me.

**Questions:**

* I found it somewhat difficult to track all the acronyms, and am wondering: why use MVRD and ExMRD (without the "v")?
* (Section 3.2) Why does the SLReviewer not fall victim to the same drawbacks of the MLLM?
* (Section 4) This may be a limitation of my own unfamiliarity with this type of work, but I did not completely follow what it means to have ExMRD without R3CoT or vice versa. Generally I need clarification on how these two contributions are distinct.

**Reviewer Confidence:**

2: The reviewer is willing to defend the evaluation, but it is likely that the reviewer did not understand parts of the paper

**Scope:**

4: The work is relevant to the Web and to the track, and is of broad interest to the community

---

### Official Review · Reviewer_rpHC · 2024-12-02

**Novelty:** 4
**Technical Quality:** 4

**Review:**

The paper introduces ExMRD, a framework for detecting and explaining rumors in micro-videos. Leveraging the Chain-of-Thought (CoT) reasoning mechanism, it comprises three steps: Refining, Retrieving, and Reasoning. These steps process and enhance video content, gather domain-specific knowledge, and generate logical explanations. To ensure efficiency, the framework uses a Small Language Reviewer (SLReviewer) to distill outputs from Multimodal Large Language Models (MLLMs).

**Questions:**

Strength:
- This paper is easy to read.
- This paper is overall completed.

Weakness:
- I am quite confused about why the baseline results of MLLMs like (GPT-4o-m, LLaVA-OV, Qwen2-VL) are reported so low in Table 2. Most of them are 50%-60%, even a 46% ACC. In my consideration, if the MLLMs are stupid to directly predict all test data as rumors or non-rumors, it will get a 50% ACC.

**Reviewer Confidence:**

3: The reviewer is confident but not certain that the evaluation is correct

**Scope:**

3: The work is somewhat relevant to the Web and to the track, and is of narrow interest to a sub-community